# Covalent Inhibitors for Neglected Diseases: An Exploration of Novel Therapeutic Options

**DOI:** 10.3390/ph16071028

**Published:** 2023-07-19

**Authors:** Erick Tavares Marcelino Alves, Filipe Gomes Pernichelle, Lucas Adriano Nascimento, Glaucio Monteiro Ferreira, Elizabeth Igne Ferreira

**Affiliations:** 1Department of Pharmacy, School of Pharmaceutical Sciences, University of Sao Paulo, Av. Prof. Lineu Prestes, 580, Butantã, São Paulo 05508-000, Brazil; erick.tavares@usp.br (E.T.M.A.);; 2Department of Clinical and Toxicological Analyses, School of Pharmaceutical Sciences, University of Sao Paulo, Av. Prof. Lineu Prestes, 580, Butantã, São Paulo 05508-000, Brazil; gmf@usp.br

**Keywords:** covalent target enzyme inhibitors, sleeping sickness, Chagas disease, Malaria

## Abstract

Neglected diseases, primarily found in tropical regions of the world, present a significant challenge for impoverished populations. Currently, there are 20 diseases considered neglected, which greatly impact the health of affected populations and result in difficult-to-control social and economic consequences. Unfortunately, for the majority of these diseases, there are few or no drugs available for patient treatment, and the few drugs that do exist often lack adequate safety and efficacy. As a result, there is a pressing need to discover and design new drugs to address these neglected diseases. This requires the identification of different targets and interactions to be studied. In recent years, there has been a growing focus on studying enzyme covalent inhibitors as a potential treatment for neglected diseases. In this review, we will explore examples of how these inhibitors have been used to target Human African Trypanosomiasis, Chagas disease, and Malaria, highlighting some of the most promising results so far. Ultimately, this review aims to inspire medicinal chemists to pursue the development of new drug candidates for these neglected diseases, and to encourage greater investment in research in this area.

## 1. Introduction

Neglected Tropical Diseases (NTD) is a group of 20 diseases affecting more than 1 billion people in the world, especially those living in poor communities of tropical areas. Caused by a variety of etiologic agents such as viruses, bacteria, and parasites, these diseases have a great impact on public health, with social and economic consequences to the affected populations [1]. These diseases are often carried by animal reservoirs, some of which are difficult to control. By affecting impoverished populations with little to no access to public health policies, NTDs do not attract enough interest for the pharmaceutical sector, due to the lack of financial profit. As a result, the chemotherapeutic arsenal is scarce for the majority of those diseases, and the lack of an efficient treatment leads to more than 170 million deaths per year [1]. Although this scenario has gradually improved, the investments keep going below the necessity. Research for new chemotherapeutic alternatives for these endemic health situations must be highly emphasized.

Surprisingly, new studies on covalent inhibitors and targeted covalent inhibitors (TCI) comprehend a potential new class of molecules with improved efficacy and security for NTDs. Compared to reversible inhibitors or mechanism-based inactivators, TCIs exhibit unique selectivity profiles. Additionally, while these previous methods are mainly limited to enzymes, the TCI approach is more versatile and can be employed for a wide range of druggable proteins [2].

TCIs can create a stable covalent bond with the target protein active site by positioning their moderately reactive electrophile warhead adjacent to a particular nucleophile amino acid residue, usually cysteine. Their pharmacological impact is sustained, while the unbound drug is quickly and continuously removed from the body from a pharmacokinetic perspective, which makes the latter less toxic [3,4]. TCIs have been employed for the development of new molecules for these three diseases over the past few years, leading to promising results and new alternatives for their treatment.

It is important to notice that the complex TCI-target is formed only after a molecular recognition of the ligand by its target molecule. TCIs are developed to not only have a reactive warhead to create a covalent bond but also must bear structural features for selective and specific interactions with its target; otherwise, no good inhibition is obtained [3]. Among all examples in this review, Structure-Activity Relationship (SAR) is studied regarding not only different bioactive warheads but also molecular modifications for a better fit.

In this review, these new inhibitors are critically evaluated regarding their usefulness for the treatment of Sleeping sickness, Chagas disease, and Malaria. Table 1 summarizes the most recent covalent inhibitors and their target in those diseases.

### Human African Trypanosomiasis (Sleeping Sickness), Chagas Disease, and Malaria

Human African Trypanosomiasis (HAT) cases are mostly found in the Democratic Republic of Congo, with fewer than 1000 cases per year. Other African countries report up to 100 cases annually, and some report sporadic cases in the last decade. Chagas disease (CD) is endemic in 21 Latin American countries, putting 75 million people at risk, with 6 million currently infected and 30,000 new cases reported yearly [93,94]. Malaria, caused by *Plasmodium* parasites transmitted by female *Anopheles* mosquitoes, is receiving more attention from the scientific community. In 2020, an estimated 241 million cases occurred worldwide, with around 600 thousand fatalities [95]. Most of the cases and deaths occurred in Africa, with 80% affecting children under five. Unfortunately, HAT and CD lack effective drugs for treating the chronic stage, and currently available treatments have multiple side effects due to their toxicity. All three diseases require urgent new therapies [93,94,95].

**Table 1 pharmaceuticals-16-01028-t001:** Targets and references containing potential covalent inhibitors towards them on three of the most important Neglected diseases.

Diseases	Targets	References
	Rhodesain (RD)	[5,6,7,8,9,10,11,12,13,14,15,16,17,18,19,20,21,22,23,24,25,26,27,28,29,30,31]
	Trypanothione Reductase (TR)	[32,33,34,35,36,37]
	Trypanothione Synthetase (TryS)	[38]
	Tryparedoxin (Tpx)	[39]
Human African Trypanosomiasis (HAT)	CATL	[40]
	Pterine Reductase I	[41,42]
	CLK1	[43,44]
	GAPDH	[30]
	Non-specific	[45,46,47,48]
	Cruzain (CZ)	[49,50,51,52,53,54,55,56,57,58,59,60,61,62,63,64,65]
Chagas Disease (CD)	Proline Racemase (PRAC)	[66,67,68,69,70]
	Trypanothione Reductase (TR)	[37,71,72]
	Non-Specific	[73,74]
	Falcipain (FP)	[15,75,76,77]
	Proteasome	[78,79,80,81,82]
Malaria	GAPDH	[4,83,84,85,86,87]
	FK506-BP	[88,89,90,91]
	Non-Specific	[92]

## 2. Covalent Inhibitors for Neglected Diseases

Numerous covalent inhibitors and their corresponding warheads are examined throughout this review. On that matter, K_i_, k2nd, k_inact_ are some of the parameters discussed and will be briefly introduced here. It is important to stress that covalent inhibition is a two-step procedure as pictured in Figure 1. Some of the constants involved in the process are described below.

k_inact_ is the rate constant defining the velocity at which the enzyme is being inactivated, being measured in s^−1^. K_i_ is defined as the concentration of inhibitors responsible for half of the maximum rate of the enzyme inhibitory activity, which is closed linked to binding affinity. k2nd is a constant frequently recommended to be used in covalent inhibitors studies but not usually applied [96]; It corresponds to k_inact_/K_i_.

This constant k_inact_ shows the rate at which the covalent bond is formed, primarily when analyzing irreversible inhibitors, and is a pseudo-first-order rate constant defining the rate of enzyme inactivation at full target occupancy [96]. Despite being a good way to understand the nature of the inhibitor, many studies correlating SAR with IC_50_ have over-relied on IC_50_, and, as recommended by Strelow, these studies “should be restricted to preliminary investigations” [96].

The correlation between k_inact_/K_i_ and IC_50_ improves as k_inact_ decreases, and the covalent term of the equation becomes less significant. To go further into the details about covalent inhibition mechanisms, see reference [96].

### 2.1. Covalent Inhibitors Studies on Human African Trypanosomiasis (HAT)

The actual therapy for HAT is based on a few available drugs: pentamidine and suramin for first-stage HAT, while second-stage HAT is treated with eflornithine, melarsoprol, and nifurtimox (Nfx); Currently, as no efficient vaccine is available, treatments have been focused on combination therapies, but, still, several adverse effects have been observed, such as melarsoprol high neurotoxicity [97].

#### 2.1.1. Oxidizing Cycle

One of the most known pathways studied against HAT is the parasite oxidizing cycle since oxidative stress plays a major role in parasite survival [98,99]. Figure 1 shows this pathway; components shown in red are the ones currently being targeted for covalent inhibition explored in this review.

##### Trypanothione Reductase (TR)

Trypanothione reductase is the most thoroughly studied enzyme of the trypanothione metabolism, as it is not expressed in mammalian hosts, and many of its inhibitors have been identified. As the parasite does not have a glutathione reductase system, which is essential for cellular redox homeostasis [32], the only structure that connects NADPH and thiol-based redox systems is TR [33].

TR is an oxidoreductase that catalyzes NADPH-dependent reduction of trypanothione (*bis*(glutathionyl)spermidine) disulfide TS_2_ to trypanothione T(SH)_2_ (Figure 2). Inhibition of this enzyme makes the parasite susceptible to oxidative stress and cellular damage.

Lu et al. studied ebselen (**1**) (Figure 2), a benzisothiazolinone derivative, anti-oxidative, and anti-inflammatory seleno-organic compound, found through high-throughput screening (HTS) for TR inhibition [34]. It presented an interesting antibacterial activity as a competitive inhibitor of bacterial tryparedoxin. They consequently examined the action of its analog, ebsulfur (**2**), against trypanosomes and TR. The results showed this molecule is active against *T. brucei*, as seen in Figure 2, with a IC_50_ of 0.092 μM in one of the strains, and showed a time-dependent inhibition of TR, indicating covalent inhibition. The inhibition mechanism discussed by the authors focuses on the thioether group present in the molecule, which covalently binds to a cysteine residue in TR.

In another approach involving TR inhibitors, Marcu et al. focused on the possible effects of chloroacetamide derivatives on inhibition of *T. brucei* TR, expanding the SAR study to tricyclic heterocycles using phenothiazine and phenoxazine [36]. Despite the lack of data, the authors claimed that compounds **3** and **4** were remarkable time-dependent inhibitors, and halogens present in their structure suggest a possible nucleophilic substitution mechanism. For more details about TR inhibition and related molecular mechanisms, read the review written by Tiwari et al. [37].

##### Trypanothione Synthetase (TryS)

Trypanothione synthetase catalyzes the synthesis of *N1,N8*-*bis*(glutathionyl)spermidine (trypanothione), the main low molecular thiol that supports many redox functionalities in trypanosomatids.

Although TryS was proved to be druggable, the design of inhibitors has been challenging due to specific characteristics of this enzyme. TryS active site accommodates four substrates and contains several regions involved in substrate binding that are highly dynamic [35].

To explore TryS inhibition, Benítez and colleagues utilized ebselen (**1**), a previously identified multi-TryS molecule [38]. The ENAMINE library(with a true-hit rate of 0.056% and 51,624 compounds) screening identified select hits that showed significant specificity against TryS. However, only ebselen (**1**) displayed a balanced antitrypanosomal and TryS inhibitory activity. The researchers conducted experiments to determine its mode of action and confirmed its covalent inhibition through the formation of Se-Cys (S) bonds, primarily through MS/MS analysis.

##### Tryparedoxin

Another step of the oxidizing cycle is catalyzed by tryparedoxin (Tpx). This enzyme is a distant member of the human thioredoxin family, and, along with T(SH)_2_, catalyzes the synthesis of compounds required for DNA synthesis, making it important for parasite replication.

A HTS investigation by Fueler et al. led to 36 compounds that inhibited the peroxiding system by ≥20% at a concentration of 40 μM [39]. As the following experiments suggested, these authors proved Tpx to be an in vivo and in vitro target and characterized compounds **6, 7** and **8** (Figure 2) as time-dependent ones through a MAL-PEG-based gel shift assay. It was also observed that the presence of the chloromethyl substituent was essential for covalent binding. Tpx Cys_40_ likely attacks the electrophilic carbon linked to the chlorine groups in a nucleophilic substitution, facilitated by some non-covalent interactions, especially hydrogen bonds (Figure 2b). A detailed study of the mechanism of these types of molecules was then performed by [100].

#### 2.1.2. Rhodesain (RD)

Rhodesain is an essential protein for parasites crossing the blood brain barrier (BBB), reaching the central nervous system. It is a key cysteine protease for the survival and infectiousness of *T. brucei rhodesiense*. It is also predominant in lysosomes, where it exerts its protease activity. RD has demonstrated to be a validated target because of its significance for the survival of *Trypanosoma* sp., and several efforts, especially those linked to covalent inhibitors, have been made in the last 10 years [5,6].

Schechter and Berger nomenclature towards sites is present in the Figure 3. Combinatorial chemistry throughout the years were applied to synthesize compounds prone to bind to the sites in several papains. That way, we are going to see similar trends in cruzain and rhodesain, for example [7].

##### Peptidic and Pseudopeptide Compounds

Kerr et al. broadened the SAR research of peptidyl vinyl Michael acceptor derivatives for RD inhibition by synthesizing a series of peptide-based compounds with the vinyl sulfones, such as K11777 [5], as lead compounds considering their structures cocrystallized with RD available in PDB [5,6]. Compound **9** presented K_inact_ of 0.00255 ± 0.00075 min^−1^, and a high k2nd of 67,000 ± 432 × 10^3^ M^−1^ min^−1^, with docking studies suggesting many hydrogen bonds with the active site (Figure 4).

In a continuation of these endeavors, Ettari et al. exploited lead compound **9**, obtaining two potent inhibitors with submicromolar EC_50_ against the parasite [8]. Based on K_i_ and k2nd values, they concluded that 3,5-difluoro and 4-CF_3_ phenyl moieties were important in P3, the residue in the side of peptide in the *N*-terminal moving further to the cleavage point (Figure 4). However, no direct association was found between enzyme activity and antitrypanosomal activity. In this context, they hypothesized that the divergence was related to enzymatic hydrolysis susceptibility and that better stability might be reached by stereochemistry alterations.

In the following year, the same group attempted to enhance the stability properties of previous compounds through the incorporation of halogens and cyclohexane in P3 and P2, the aminoacid residue of the peptide in the *N*-terminal moving further to the cleavage point respectively [9]. As a result, two compounds were shown to be interesting hits, with highlight being **14** (Figure 4) of k2nd 52,200 ± 26,200 M^−1^ min^−1^ and K_inact_ of 0.0006 ± 0.0001 min^−1^, which balanced interesting antitrypanosomal and inhibitory activity.

The presence of bulkier groups in P1 residue, the immediate N-terminal to the cleavage site, was discouraged for further experiments as it demonstrated lack of affinity towards RD in P1′, the immediate C-terminal to the cleavage site. This led Previti et al. to enlarge the SAR and perform alterations in P3, especially [10]. In P1 and P2, hPhe and Phe were maintained, respectively, as they showed good activity, and a variety of groups were tested in P3. The peptide bond was substituted by a urea bond, an amide bioisoster, to provide more stability. The compound that presented the best inhibitory activity towards RD did not have good antitrypanosomal activity. These results suggested that aromatic substituents in P3 were essential to modulate properties concerning cell permeability. The compound which best balanced antitrypanosomal and RD inhibition activity was **15** (Figure 5), with a K_i_ of 0.51 ± 0.36 nM and EC_50_ of 1.65 ± 0.07 μM. The presence of chlorine and methyl probably improved cell permeability, thus reaching interesting antitrypanosomal activity.

Most recently, Previti et al. performed docking experiments with **10** (Figure 5) and hypothesized that the carbonyl in P3 did not perform any major non-covalent interactions, leading to investigation of alternative substituents such as amines and a vinyl ester warhead in P1 [11]. As a result, they obtained 3 interesting hits towards RD, **12, 13** and **14**. The presence of vinyl ester instead of vinyl ketone in P3 resulted in a loss of affinity (**10**, 1.1 pM → **12**, 5.8 nM). Compound **14** presented the best antitrypanosomal activity, EC_50_ = 7.0 ± 1 μM, and the docking experiment (Figure 5) revealed its possible mode of interaction.

On the other hand, Chio et al. [12] kept the homophenylalanine (hPhe) at the P1 site unchanged, as it has been shown to be a critical structural requirement for affinity towards the target enzyme. Following the methodology of a previous study [13], they designed a series of aza-nitrile compounds [14] where Phe and Leu residues, typically preferred by rhodesain, were sampled at the P2 position. The amino group of the P2 substituent was protected with a fluoro-benzoyl group, which spanned into the P3 region to optimize interactions with the S3 pocket. All the compounds showed activity in the range of 16–122 nM. Compound **13** and **14** were the most promising, as the former showed interesting antitrypanosomal activity, likely due to its high lipophilicity compared to other compounds in the series, and the latter displayed the best activity against rhodesain, as well as interesting antitrypanosomal activity (Figure 4). These compounds represent new lead compounds for further investigation.

In another approach, Ettari et al. designed RD inhibitors containing a 3-bromoisoxaline moiety as a warhead and a peptidomimetic scaffold as a recognition moiety towards RD [17]. Compounds with interesting activity were obtained from the lead compounds **17** and **18**, highlighting the compound **19** with K_i_ (RD) of 0.96 μM, inferring that 3-bromoisoxaline might be a good warhead when coupled with an interesting recognition motif (Figure 6).

Following the work of [17], Chio et al. synthesized a series of compounds containing a vinyl ester as a warhead in place of 3-bromoisoxaline [13]. Among the compounds, **20** (Figure 6) was the most active against both RD and the parasite, providing a good lead for further work. Because of its high lipophilicity, this compound was hypothesized to cross biological membranes more effectively, which possibly contributed to its high activity.

Also in a continuation of the study developed in [17], Royo et al. used **21** as a lead compound and envisioned similar dipeptidyl enoate inhibitors bearing a carbonyl group instead of a sulfonyl one and an alpha carbon group in P1 [18]. Compound **22** was then obtained and shown to be a potent inhibitor of RD, with IC_50_ of 16.4 ± 3.7 nM and k_inact_/K_i_ of 1,610,000 ± 297,000 min^−1^μM^−1^. The researchers performed docking experiments to confirm that compounds **21** and **22** had a similar conformation, in which their warheads are positioned close to Cys_25_. The ester group of **22** possibly interacted with Gln_19_, Trp_184_, and His_162_, similar to the sulfone group of **21**. The researchers then tested these dipeptidyl enoates against other cysteine proteases besides RD, such as falcipain and cruzain. Their goal was to optimize interactions at S1, S2, S3, and S1′ [19]. Compound **24** showed the best inhibitory activity against *T. brucei*. In vitro assays revealed that the IC_50_ values of **22** and **24** varied between 0.3 and 9.5 μM. Compound **24** was also active against *P. falciparum*, with an IC_50_ of 6.81 μM [19] (Figure 7).

Using a different approach, Ettari et al. tackled the high cLogP and molecular weight from previous leads, especially **25**, and synthesized one compound in the low micromolar range activity, **26** [20], being interesting also due to its minor cytotoxicity towards mammalian cells (TC_50_ > 100 μM), besides presenting an IC_50_ of 5.3 ± 0.03 µM. Concomitantly, the same group explored the benzodiazepine scaffold linked to bromoisoxaline as a warhead, as employed by [17], maintaining the adamantyl group (Figure 7) [21]. They could then keep activity (**27**, K_i_ = 1.26 μM, Figure 8) similar to the lead compound and assert the importance of the adamantyl group for the modulation of physicochemical characteristics and, consequently, for trespassing the parasite membrane [21].

In order to optimize compound **21** (Figure 8) [22] and to improve RD inhibition and selectivity toward cathepsins L and B, Jung et al. explored SAR changing the recognition moiety and substitution pattern near the warhead [23]. As a result, aromatic insertions on P3 were interesting for inhibition against RD (**30**, K_i_ = 12 nM). As for the selectivity, the incorporation of 4-methyl group in the phenyl ring improved selectivity in comparison with human cathepsins (**37**, K_i_ = 8 nM, 12× CatL and 200× CatB). Also, it was noted that **29** showed interesting potency towards RD with a dissociation constant of the initial encounter complex K_i_ = 24 nM, and with high affinity complex formed in the second step (K_i_* = 3 nM) (as Jung et al. defines it, “Ki* denotes the dissociation constant of the high-affinity complex in the case of biphasic, time-dependent inhibition”). Docking studies also suggested this complex to be stable (Figure 9).

As for the last highlight, they compared compounds **28** and **29** as they only differ by the presence of fluor (Figure 8). For its absence, **29** proved to be an irreversible covalent inhibitor.

Finally, Chenna et al. described a series of reversible time-dependent inhibitors of cruzain-based on compound **21** and GSK2793660 (compound **33**) [24]. These two compounds, especially due to its ability to form irreversible covalent bonds (Figure 10), presented toxicity issues. These compounds, **21** and **33**, led the authors to synthesize compounds with less electrophilic moieties, such as 2-vinyl-pyrimidine with K_i_* = 0.1–0.4 μM. Soon after, docking studies suggested that heterocycle inhibitors might perform Michael reversible addition to Cys_25_, such as the example shown for compound **29** in Figure 11.

Among the hit compounds, **31** presented an interesting EC_50_ in *T. brucei* of 5.8 μM, similar to what was pointed out from **24** previous kinetics studies. Compound **32** also deserved a highlight due to its values of EC_50_ being widely correlated with K_i_*, what is a good indication of the correlation of activity with covalent inhibition (Figure 8) [24].

##### Non-Peptidic Compounds

In recent years, RD inhibition has been also explored with non-peptidic compounds as an attempt to improve pharmacokinetics characteristics in comparison to what was obtained with peptidic compounds [25].

McShan et al. used the irreversible tethering method to explore a library of 200 electrophilic fragments that could potentially lead to covalent inhibitors [25]. In contrast to peptidic ones, compounds **34** and **35** were found to be non-peptidic covalent inhibitors of papain and might possess interesting pharmacokinetic properties. Additionally, these substances had low cytotoxicity and low micromolar activity against *T. brucei* (Figure 11). The authors also stated a possible mechanism of interaction between the covalent fragments studied and the target [26], which is depicted in Figure 12.

In another approach, Ehmke et al. explored the SAR of aryl nitriles as covalent inhibitors of RD and hCatL, human analogous to RD [26]. They hypothesized that the toxicity of the compounds was directly associated with their warhead electrophilicity and used DFT calculations to understand why some electrophilic moieties had higher inactivation rates than others. Finally, two compounds (**38** and **39**) with a single digit nanomolar activity were obtained towards RD (9 nM and 2 nM, respectively), showing low selectivity towards the human analogue.

In an attempt to improve the selectivity of these both compounds, Ehmke et al. also constructed another series of compounds, varying P1, P2 and P3 (moieties linked to S1, S2 and S3 as shown in Figure 13) [27]. One of the key findings of the study was that variations in the S1 region did not have a direct impact on inhibitory potency. However, the S2 region was found to be the primary contributor to binding affinity. In particular, compounds with cyclohexyl-based substituents, such as compound **39** shown in Figure 13, showed the most promise. The study also revealed that the S3 region preferred aromatic vectors, likely due to π-stacking interactions. Compounds **38** and **39**, with K_i_ (RD) = 3 nM, were found to be the best candidates when varying the S3 region. Of the two, compound **39** demonstrated greater drug potential due to its lower activity against hCatL (K_i_ = 32 nM), indicating better selectivity.

In a continuation of studies from [27] and observing the little influence P1 had on activity and affinity towards rhodesain, Schirmeister et al. published a literature review on cathepsin L inhibitors in an attempt to expand the SAR of RD inhibitors focusing in the S2 and S3 regions (Figure 14) [22].

The preference of RD for hydrophobic residues was corroborated with 3- and 4-Br-Phe residues presenting interesting activities. The authors also noted that the compound with a triaryl extremity showed interesting inhibition against RD (**41**, Figure 13, K_i_ = 5.3 nM) and a selectivity index of 200 versus cathepsin L.

Crystallographic experiments were also performed in addition with compound **41**. These led to the discovery of some hydrophobic interactions next to S2, similar to what was previously proposed [27] (Figure 13), hydrogen bonds, such as between the ligand and Gly_66_ backbone, and the covalent bond between sulfur from Cys_25_ and the warhead (Figure 15).

Although compound **41** inhibitory activity against the enzyme was at nanomolar range, the same was not translated into antitrypanosomal activity shown by ATPlite assay (EC_50_ = 8 μM), being less significant than what was observed previously [27]. One possible explanation was the lack of basic groups in **41**, shown to be a good characteristic of those compounds that presented better antitrypanosomal activity [22,27].

Using a library of 49 hCatL inhibitors, Giroud et al. performed a phenotypic screen to identify potential hits against RD. In this context, a macrocyclic lactam presented an activity of 11 nM in K_i_ for RD (compound **42**, Figure 15) [28]. Soon after, the same group simplified the molecule and obtained two compounds with similar activities (**43** and **44**) at 7.4 nM and 93 nM, respectively [29]. Even though the activity was lower than in previous studies, these compounds were more selective against hCatL. While the first study pointed out a K_i_ of 10 nM for this enzyme, compounds **43** and **44** presented 0.6 and 0.2 nM, respectively, showing a good potential for this kind of structure.Compound **45** which remarkably inhibited cell growth of *T. b. rhodesiense* (IC_50_ (48 h) = 64 nM) and had its structure evaluated towards RD (Figure 16).

#### 2.1.3. Other Compounds and Targets Related to HAT

##### Pterine Reductase 1

Tetrahydrobiopterin [H4B] is a necessary cofactor for enzymes such as aromatic amino acid hydroxylases to catalyze different hydroxylation processes. H4B can be generated from GTP or rescued from dihydrobiopterin [H2B] by NADPH-dependent dihydrofolate reductase in humans or NAD[P]H-dependent 6,7-dihydropteridine reductase which regenerates quinonoid dihydrobiopterin, a byproduct of hydroxylation processes. Biological and genetic evidence suggests that trypanosomatids lack the ability to synthesize pterins de novo, and are totally dependent on the uptake of extracellular pterins for growth [41]. Pterin reductase 1 [PTR1] is an enzyme that reduces dihydrobiopterin in *Trypanosoma* spp., being essential for its survival.

Over the past 10 years, papers have been published, in which pyrimidine-based compounds have been exploited for protein kinase inhibition, topoisomerase inhibition, antibacterial, anti-inflammatory and antiparasitic activity, and dipeptidyl peptidase IV inhibition. Besides, pyrrolopyrimidines had the advantage of carrying a pharmacophore similar to the recognition motif of the parasite P2 aminopurine transporter.

Khalaf et al. considered 2,4-diaminopyrimidines to be important substructures at the outset of this work [42]. Recognizing that physicochemical properties also play an important role in the biological activity of substituted pyrimidines, they investigated 4-alkoxy and 4-alkylamino substituents. Also, this requirement for transport into trypanosomes and the possibility that the specific transporters prefer 4-aminopyridine to engage the hydrophobic pockets of PTR1 evident from crystallographic studies, 5-alkyl, 5-aryl, 6-alkyl, and 6-aryl pyrrolopyrimidines together with 5,6-disubstituted compounds were all studied (Figure 17a). Even though compounds **46** and **47** did not exhibit the best enzyme and in vitro activity, they might be investigated further for the inclusion of a formaldehyde motif, which could potentially form a reversible covalent bond to Cys_168_.

##### Kinase CLK1

CLK1 is a crucial protein for mitosis in *Trypanosoma* spp., and is a target for the antibiotic hypothemycin, highlighting its potential as a drug target [44]. In their research, Saldivia and colleagues found a link between the inhibition of recombinant TbCLK1 and the death of the parasite [43]. Only compounds containing Michael acceptors were studied, leading to the identification of compound **48**. To establish the significance of the Michael acceptor for inhibition activity, they synthesized **49** without the unsaturated bond, which in turn abolished compound activity (Figure 17b). Crystallographic analysis revealed that Cys_215_ forms a covalent bond with the Michael acceptor, while Asp_218_ appears to be involved in an ionic bond with an amine. Additionally, the nitrogen from Pro_214_ is close to the benzyl linked to pyrrole, and Tyr_212_ appears to form hydrogen bonds with oxygen through its backbone (Figure 17c).

##### Cathepsin-like (TbCATL)

Two cathepsin-like proteins are of paramount importance inside the parasite: TbCATB and TbCATL. In recent studies, however, only TbCATL showed to be essential for the parasite [40]. Compound **50** selectively inhibited the activity of TbCATL in trypanosomes without affecting TbCATB. This molecule showed a dose-dependent inhibitory effect with an (IC_50_) of 0.39 µM, indicating prominent inhibition of TbCATL (Figure 18d). Steverding et al., lastly, stated that TbCATL is essential for the survival of bloodstream forms of *T. brucei* and this suggests that future drug development programs shall focus on the rational design of TbCATL inhibitors [40].

##### Glyceraldehyde-3-Phosphate Dehydrogenase [GAPDH] and TR

Considering the design of new drugs, molecular hybridization has been gaining importance in the last few years. Bellutti et al. aimed to modulate two vital targets in *T. brucei* and *T. cruzi*, TR and GAPDH [30]. For that, they considered two different lead compounds and synthesized a hybrid, leading to 10 different derivatives (One of the lead is compound **51**, Figure 18e). An interesting feature present is the amine, holding a positive charge at physiological pH, which would present interaction with a pocket filled with negative residues. Compound **53** covalently inhibits the GAPDH through Cys_166_ and presented the best activity against the two enzymes with an IC_50_ (GAPDH) of 5.3 μM and K_i_ (TR) of 2.32 μM. It is worth mentioning though that this trend was not corroborated by in vivo activity, being one of the weakest inhibitors of the series [46] the best one considering in vivo activity, which would suggest off-target activity.

#### 2.1.4. Natural Compounds and Derivatives

The natural world is certainly a good source to start searching for new possible drug candidates [45]. In an isolated study, Oli et al. took advantage of the great structural variety of marine sponges and the fact that some of their extracts were tested against cysteine proteases [46]. From bioactivity-guided fractionation of the crude cyclohexane extract, they obtained plakortide E [47] and found out it had a IC_50_ of 5 μM in *T. brucei* and a time dependent IC_50_ towards RD, varying from 257 μM in 5 min to 77 μM in 60 min, which is indicative for a covalent inhibition given the presence of Michael acceptor.

Considering various compounds previously reported for trypanocidal activity, as well as the recent discovery of furanoheliangolides with high antitrypanosomal activity, the interest in such compounds has increased indeed. Lenz et al. showed that the natural 4,15-isoatriplicolide type sesquiterpene lactones inhibited trypanothione reductase; particularly, 4,15-isoatriplicolide tiglate exhibited exceptionally potent activity against *T. brucei rhodesiense* with an IC_50_ value of 15 nM [48]. In 2019, this same group synthesized compound **55**, which also proved to be an interesting compound, presenting TbTR inhibitory activity of more than 10% in only 15 min.

Zhang et al., on the other hand, approached natural phenyl vinyl sulfone compounds-derivatives [31]. They observed compound **56** to be inactive against RD at 20 µM after 1 h incubation, K_inact_/K_i_ of 99 M^−1^ s^−1^ and to show reasonable in vitro activity, with an IC_50_ of 5.97 µM. Another interesting feature was its inactivity towards human analogous cathepsin L. Cys_25_ was discovered to be adjacent to the vinyl moiety in docking investigations. The homophenyl is near the P1 site, while the quinoline is next to P2. The phenyl moiety was predicted to interact with Gln_19_ and His_162_, while quinolyl had hydrophobic interactions with Met_68_. On the other hand, P3 did not have a moiety of its own. Thus, regarding these experiments, it is of interest to highlight the synthesis of compound **57**, which presented interesting in vivo antitrypanosomal activity, and **58**, which presented a great selectivity index of over 159 (Figure 19).

### 2.2. Covalent Inhibitors Studies on Chagas Disease [CD]

#### 2.2.1. Cruzain [CZ]

Cruzain is the largest cysteine protease from *T. cruzi*, and it is associated with many crucial biological processes inside the parasite, such as differentiation, invasion, and proliferation in host cells [49]. The sites most explored regarding drug discovery and this enzyme are S1, S2, and S3, as shown in Figure 19.

##### Peptidomimetics

One of the major efforts in assessing drug design against CD and CZ inhibition involves peptidomimetic compounds. The use of these structures comes from the fact that cysteine proteases target protein-like structures. CZ, as a cysteine protease per se, is not different, and using this scaffold as a lead has allowed researchers to design many potential inhibitors [50].

Avelar et al. used a molecule established by [51] as a lead (**59**, Figure 20), and explored nitrile as a warhead for TCI against CZ [52]. After many modifications in P1, P2 and P3 (as seen in Figure 18), they synthesized a derivative containing 1-methyl-3-tert-butyl pyrazole in P3 and chlorine in meta for P2 (compound **63**, pK_i_ = 7.2). Furthermore, crystallographic experiments revealed that phenylalanine did not penetrate as effectively as *m*-chlorine in pocket S2. The authors determined that efficient penetration of S2 was only achievable when 3-chloro and 1-methyl-3-pyrazole groups were present. Despite finding this information, they were unable to demonstrate a substantial structure-activity link for amastigote forms.

Compounds **61** and **62** were the two most active compounds against CZ synthesized by the aforementioned group, with pK_i_ equal to 7.8 ± 0.03 and 7.6 ± 0.02, respectively (Figure 20).

The mechanism of action concerning the molecules cited is thoroughly explored by Dos Santos et al. [55]. Experimental data has pointed out the covalent, reversible inhibition of nitrile-based compounds, depending on Cys_25_ and His_162_ residues. The imidazole group in His_162_ polarizes the SH group of Cys_25_ and enables its deprotonation, promoting Michael substitution on the ligand (Figure 21).

Through further efforts related to Avelar et al. [52], Lameiro et al. attempted to optimize compound **60** to achieve metabolic stability by substituting amide for trifluoro groups [50]. The authors also explored the exchange of Phe by Leu at P2, the stereochemistry of adjacent CF_3_ carbon, and how the bioisosteric replacement H/F affected activity, besides exploring selectivity towards various Cysteine proteases.

On the other hand, Cianni et al. used target-based molecular design to construct a series of potential CZ inhibitors based on compound **58** obtained by Avelar et al. [53]. They then hypothesized that Asp_16_, Gly_66_ (S2), and His_162_ (S1) were critical amino acids for CZ inhibition based on PDB data (PDB code: 1ME3). Furthermore, S3 had not been thoroughly explored before their work. In this regard, they used compound **58** [52] as a prototype and searched for it in silico, kinetics, and thermodynamics experiments to evaluate many derived analogs. The researchers determined that Ser_61_ interacted better with chlorine atoms in *meta*, which might explain halogen interactions observed through molecular dynamics. To further understand this pattern, the authors examined different halogens in *meta* position and discovered that fluorine allowed for a more violent rotation of the molecule and proposed a matched pair molecular analysis, focusing on modifications in P3. The most significant change observed was from *m*-H to *m*-I, followed by *m*-H to *m*-Cl, as corroborated by molecular modeling experiments. In addition, they mentioned bond donors were less potent than -Cl, which indicates the existence of halogen bonds.

The *m*-chlorophenyl, *m*-bromophenyl, and *m*-hydroxyphenyl moieties presented the correct orientation inside P3 to interact with the oxygen of the Ser_61_ residue, according to molecular dynamics studies. To properly examine this pattern, they synthesized 11 novel CZ inhibitors with various substituents in P3. The results were in good agreement with the respective molecular dynamics simulations, once the best compounds were precisely those with a lower fluctuation of the P3 ring [52] (Figure 22). The substitution of a hydrogen atom in *meta* position in P3 for an iodine atom increased the pK_i_ by 0.8 log units.

Alves et al., following Avelar et al. study [52], verified the interaction capacity of peptoids to target cysteine proteases [54]. An interesting CZ inhibitor (**66**, pK_i_ = 6.8) was identified, although not as potent as the analog synthesized by [1,51] (pK_i_ = 7.2). Furthermore, docking experiments suggested that, except for benzyl, the S1 pocket of CZ might accept the bulk of the investigated groups [52]. In S3, there is a fit for biphenyl groups with an interesting proximity between the second ring of two hydrophilic residues Asn_70_ and Ser_61_, characterizing probably hydrogen-π interactions (Figure 22).

In another approach concerning nitrile warheads, Burtoloso et al. explored odanacatib (compound **67**), a classical cathepsin K inhibitor, as a lead. CZ inhibitors were successfully synthesized, albeit this activity was not translated into in vitro effectiveness. In addition to the observations on cathepsin K inhibitors based on cyclohexane dicarboxamide and odanacatib, scaffolds being bound to their target similarly, three interesting compounds were obtained, with **68** reaching a pEC_50_ of 6.9, and **68** and **69** reaching a pK_i_ of 5.6, and **70** a near pKi of 5.5 (Figure 23). Compounds **68**, **69** and **70** showed higher levels of antitrypanosomal activity than expected for CZ inhibitors. As Burtoloso et al. studied, potency toward CZ tends to increase with molecular size, despite **68** being an exception to this trend [56]. Compound **68** and its *R*-enantiomer have a contrasting activity probably due to mirrored orientation of S3 moieties. Although the compounds synthesized derived from a covalent inhibitor and present warheads were proved to be potentially covalent in the past, no study was performed from this perspective, but we thought to be worth mentioning.

In the search for new compounds, new drug discovery methodologies have been developed and used. Gomes et al., for example, explored some prototypes in which the substitution of biphenyl for pyrimidine in P3 caused a change in selectivity towards hCatL based on the previous works [57]. Also, when biphenyl was substituted for phenyl, 0.4 log units of activity were lost (Figure 23). Furthermore, they investigated how inversion of configuration in P3 and replacement of pyridyl with benzyl were not additive, and further cellular testing did not allow any further conclusion. On the other hand, the substitution of cyclopropyl for benzyl increased the activity. A configuration inversion in P3 of the carbons also caused a change in affinity (6.7 for *R* and 7.4 for *S*). Among all the synthesized compounds, **74** presented a pK_i_ of 9.2 and an interesting selectivity when compared to Cathepsin L (Δpk_i_ = 3.4). As they observed, P3 accommodated and proved tolerable to phenyl and biphenyl groups and P1 benzyl. The replacement of phenyl by biphenyl at P2 caused an increase in pK_i_, while the replacement of cyclopropane by benzyl methylene increased pK_i_ for the *S* configuration, but decreased it for the *R* enantiomer. Regarding P2, the substitution of Leu for Phe decreased the affinity in some cases, and the substitution in P3 by halogenated compounds proved to be tolerable. Nevertheless, it is important to note how this kind of compound could be translated into therapeutics.

Nitrile-containing compounds proved to be bioavailable [59]. The authors used the compounds synthesized by Beaulieu et al. to obtain in vivo and in vitro activities, observing a great correlation between enzyme potency and antiparasitic activity [58]. The most remarkable results were shown compounds cz007 (**75**) and cz008 (**76**) (Figure 24). In another approach using nitriles as warheads, Silva et al. performed a comparison study among different warheads [60]. This study had the objective of evaluating warheads based on a carbon-nitrogen double bond. The reference compound for the study and the compounds obtained by lead optimization are depicted in Figure 24.

Silva et al. studied vinyl sulfones, generally irreversible covalent inhibitors, and vinyl amides and esters were synthesized to test whether the replacement of sulfonyl with a less strongly electron-withdrawing group would lead to a reversible inhibition of CZ [60]. The vinyl ester **72** (k_inact_/K_i_ = 9.8 × 10^3^ s^−1^M^−1^) (Figure 23) was the compound tested for irreversible covalent mechanism, although it was less active than K-777 (**79**) (k_inact_/K_i_ = 5.1 × 10^5^ s^−1^ M^−1^). Vinyl esters and amides did not offer any advantage towards more potent nitriles once they were of higher molecular weight and would also be associated with poor pharmacokinetics. In this context, aza-peptide nitriles have been shown to be stable inhibitors of papain-like cysteine proteases and appeared to be useful in activity-based problem regarding *T. brucei* in the past [14,61] Compound **77** (Figure 24) has a pKi of 8.7 ± 0.02 (Figure 24). This compound is interesting once it lacks the hydrogen bond donor in P1 of typical CZ inhibitors. Compound **77** had the best results in cell-based assays, with a pEC_50_ of 5.3. However, it also presented the higher pCC_50_ values 5.0. Azapeptide and aldehyde can be more potent than the traditional dipeptide nitriles.

With the purpose of explaining why some inhibitors are reversible or irreversible, Silva et al. performed quantum mechanics/molecular mechanics studies in addition to a synthesis of the **79** analog, Neq0682 (**80**) (Figure 25) [62]. QM/MM analysis suggested that His_162_ plays a significant role as a polarizing residue, making it easier for Cys_25_ to attack the triple bond, being the energetic barrier very similar for both, **79** and **80** (Figure 25) which is very similar to previously shown in Figure 20. The explanation for the reversibility of the process is that the energetic barrier from the **79**-CZ complex is bigger than the **80** one, which was demonstrated by the higher negative value of ΔG in the first reaction.

##### Non-Pepditic Compounds

Non-peptide compounds have been also studied by Jimenez et al. They explored *N*-acyhydrazones (NAH) and benzenesulfonyl (BS) as groups with potential antitrypanosomal activity totaling 14 compounds designed [63], obtaining compounds with activity, having in common the presence of strong electron-withdrawing groups in *para*-position on the benzene ring.

Maldonado et al. substituted the sulfonyl by carbonyl group to obtain new *N*-propionyl *N***′**-benzeneacylhydrazone derivatives (**83** and **84**) and evaluated their trypanocidal activity (Figure 26) [64]. As a result, compound **83** presented an interesting activity against NINOA strain, and compound **84** presented LC_50_ comparable to Bzn.

##### Natural Compounds

Silva et al. explored naphthoquinone analogs known to exhibit many biological properties, such as anticancer and antitrypanosomal [65]. In this work, they performed a virtual screening on CZ and RD. Thus, 14 molecules were identified as potentially active and were further synthesized. Compound **87** (Figure 27) was a hit compound known as a time-dependent inhibitor, and active in CZ and RD as compounds **88** and **89**.

After performing SAR analysis to identify the most promising molecular features of the compounds, they observed through reversibility assay by dilution that only **87** exhibited covalent inhibitory properties, so its mechanism was investigated using covalent molecular docking and molecular dynamics. Therefore, this compound was studied as possible covalent inhibitors of RD. According to the results of the molecular dynamics, **87** established contact with His_129_ and Cys_25_. MM-PBSA calculations revealed that van der Waals forces are the most significant to stabilize it. Compound **87** has also been shown to have a strong affinity for RD, which is supported by its low IC_50_ value. This compound might undergo a nucleophilic attack by Cys_25_ residue through a C3-Michael addition, as demonstrated by covalent molecular docking and DFT simulations. To assess the reversibility of such molecules, they performed a dilution experiment in which they observed the remaining enzyme inhibition upon dilution for a known irreversible cysteine protease inhibitor E-64, but the same was not observed for **87**.

In a different approach, Boudreau et al. chose to investigate analogues and derivatives of gallinamide A (compound **90**), an isolated cyanobacteria compound [73]. In this instance, one of the compounds was shown to have a LD_50_ of 5.1 ± 1.4 nM, while compound **90** had a LD_50_ of 14.7 nM against *Trypanosoma cruzi*. Following this, Silva et al. found compounds **92**, **93** and **94**, whose activities exceeded **90** by 15 times [74]. The structural difference between **92** and **93,** in comparison with **94,** was the lack of CH_2_-CH_2_-Ph in P1, which excludes the stabilizer effect of intramolecular interactions between indole and CH_2_-CH_2_-Ph. They also performed calculations on per-residue free energy decomposition-structure ensemble.

Similarly to [73], Silva et al. approached the CZ inhibitory potential of these compounds [74]. As a result, they explored P4 to module pharmacokinetics properties. Simulations of the interaction of **94** with CZ showed that indole in this compound can occur in two principal conformations: interacting with Trp_184_ and establishing it, and stacked with CH_2_-CH_2_-Ph in P1. In the cocrystal, **92** and **94** presented similar activity toward CZ, which might suggest a gain in the binding enthalpy due to intermolecular interactions of the indole ring in **93**, comparable to the loss of entropy associated with intermolecular interactions of the indole ring in **94**.

#### 2.2.2. Other Targets Related to CD

##### Proline Racemase [PRAC]

Proline racemase catalyzes the interconversion of L- and D-proline enantiomers [66] with two different residues directly involved in isomerization: Cys_130_ and Cys_270_ [67]. This is important for several processes in cell function, including *T. cruzi* wall construction and immune escape. Also, for its presence in every single stage of T. cruzi cycle and since the parasite is no longer viable after knocking this gene down and gets more virulent if it is overexpressed, this target can be promising [68].

After identifying several isoforms of *T. cruzi* proline racemases, Berneman et al. chose 2-pyrrolecarboxylic acid (PYC) as a starting point as it is a known PRAC inhibitor with a deficit in water solubility [67]. This latter feature was the focus of the group and led to the synthesis of several pyrazole inhibitors that showed to be more soluble, but that did not present a good affinity for PRAC. In another approach, they performed virtual screening experiments and constructed a QSAR model, leading them to two novel, poorly reactive Michael acceptor compounds: OxoPa (**95)** and BrOxoPA (**96**) (Figure 28).

A few years later, Amaral et al. performed crystallographic studies of the two aforementioned compounds [69]. From this data, they proposed new molecular modifications, synthesizing an optimized molecule (NG-P27—**97**). Unlike **95** and **96**, **97** displayed effects on parasite cultures. Both compounds, **95** and **96,** exhibited dose-dependent trypanocidal activity for the CL and Y parasite strains. While multiple additions of benznidazole [three times] increased parasite growth inhibition, no cumulative effect was observed after single or multiple treatments with **97** by Amaral et al. [69]. The same group carried out crystallographic studies, and indicated the proximity of the cysteine group towards the double bond in compound **97** (2 Å, approximately) (Figure 28).

Melo et al. expanded the study and synthesized three other derivatives of **97** in order to optimize some pharmacokinetic features for testing against parasites [70]. These compounds had no inhibitory activity against other cysteine proteases. Among these compounds, **98** (Figure 28) was chosen for in vivo assays and had more efficacy in live parasites than the reference compound Bzn.

##### Trypanothione Reductase (TR)

TR is an essential enzyme for the trypanothione-based thiol metabolism, as stated before for *Trypanosoma brucei*.

Beig et al. employed in vitro and in silico drug discovery methodologies, and 82 novel inhibitors showed activity in the range of nanomolar against *T. cruzi* TR (Figure 29) [71]. The best three hits were also relatively active in cell culture (IC_50_ < 2 μM, comparable to Nfx) and shared an aromatic core ring. Two hits (**101** and **102**) are shown in Figure 30, and the authors performed experiments that suggested these molecules might act as reversible inhibitors of TR. More details about TR and the structural insights of this protein are described in the reviews written by Tiwari et al. [37] and Battista et al. [72].

### 2.3. Malaria

*P. falciparum* has developed resistance to almost all antimalarial drugs used in prevention and treatment. Although most cases are in underdeveloped countries, globalization has made it possible for malaria to spread specially when left uncontrolled [95].

#### 2.3.1. PfGAPDH

GAPDH catalyzes the conversion of glyceraldehyde 3-phosphate (G3P) into 1,3-biphosphoglycerate (1,3-BPG), which is the sixth stage of glycolysis, reducing NAD+ to NAD through a catalytic Cys_153_. Its potential as a parasitic target comes from the fact that, mainly at the amastigote stage, glycolysis makes up every amount of energy storage for the parasite [83].

PfGADPH is inactivated by compounds capable of alkylating thiols such as 3-bromo-isoxazolines. In this regard, Bruno and coworker decided to analyze the potential activity of avicin and Br-avicin, as well as some of their analogs, and how strong and effective bromine warheads were for PfGADPH inhibition [83]. From their series, two compounds presented the best results: **103** and **104** (Figure 30), with the latter presenting the best covalent parameters (K_inact_/K_i_ = 10.7 ± 2.8). Docking experiments suggested that the isoxazoline moiety, which was near His_180_ and Tyr_323_, probably interacted with NAD+ through π-stacking interactions (Figure 31). Furthermore, carbonyl oxygen carried out another hydrogen interaction with a ribose hydroxyl group, while the amine group is over sugar without any other contact. Cys_153_ was responsible for the nucleophilic attack, later confirmed by UV-vis spectrophotometric detection. It is worth to note that while the glycolytic function of PfGAPDH is inhibited by heme binding this function in enzyme from mammalian is not, which is indicative of difference in the catalytic center [84,85].

In an effort to further investigate this area, Cullia and colleagues expanded on the previous SAR study [4]. They improved upon earlier compounds by modifying the amino acid component, inspired by a group of L-glutamine-dependent amidotransferase inhibitors [83]. Time- and concentration-dependent experiments were conducted with PfGADPH inhibitors, and compound **103** (Figure 30 and Figure 31) showed the most promising results for inhibiting the protein. The authors attributed this to the presence of a minor substituent linked to the ester group, which improved the compound orientation towards Cys_153_ from either the Pi or Ps site. In addition, the amino group interacted with Thr_154_ and Gly_215_ through hydrogen bonds, while the methyl ester group imitated the location of the phosphate substrate in P1 (Figure 31). This orientation facilitated bromine release and subsequent enzyme inactivation. However, this trend of activity was not observed in antiparasitic activity, which the authors suggested was due to the essential role of the α-carbonyl group in transporter recognition. The authors also highlighted compounds **105** and **106**, which showed lower activity than **103** but exhibited considerably lower toxicity, with selectivity indexes greater than 175 and 357, respectively.

Galbiati and colleagues aimed to improve the metabolic stability and toxicity of earlier compounds containing ester and amide moieties, which hydrolyze rapidly into carboxylic acid in biological media [86]. They substituted the ester and amide groups with bioisosteres such as oxadiazole, triazole, oxazole, thiazole, and thiadiazole and evaluated GAPDH inhibition. Compound **107** was found to be the most promising of the series, Achieving submicromolar activity for both strains of *P. falciparum*. These new compounds also had higher lipophilicity, which may contribute to increased membrane penetration compared to the initial compounds, with logD values of 1.41 for compound **105**, 0.76 for compound **106**, and 3.35 for compound **107**. The authors stressed the synthetic simplicity of these novel compounds and suggested that further studies, including with PfGAPDH itself, could be conducted. For information on other types of compounds and inhibitors related to 3-halo-4,5-dihydroisoxazole, refer to [87].

#### 2.3.2. Falcipains

There are four papain-like cysteine proteases characterized for *P. falciparum* named falcipains, FP-1, FP-2 and FP-2′, FP-3. FP-2 and FP-3 diverge by 37% in amino acid sequence and are expressed in the acidic vacuoles of schizonts and trophozoites. Studies have shown that inhibiting these two latter enzymes prevents hemoglobin hydrolysis and parasite development. Also, knocked-out experiments concerning FP-2 led to the intermittent hindrance of hemoglobin hydrolysis in trophozoites, which made these parasites susceptible to inhibitors, although metabolic recovery was observed due to the probable expression of FP-3 [75].

In light of the studies of these targets, Ettari et al. have conducted a research on the construction of new peptidomimetics with a 1,4-benzodiazepine scaffold introduced into a dipeptide moiety, simulating a sequence of peptides necessary to bind mainly with FP-2 (Figure 32) [75].

From previous studies, Ettari et al. studied the effects of modifications in some parts of the structures present in **108 [76]**. The methyl group at P1′ was replaced, which maintained irreversible inhibition but significantly reduced affinity and inhibitory potency. This difference was attributed to the divergence in lipophilicity of the compounds. Docking experiments provided further explanation, and the k_2_ values displayed by the compounds indicated a disfavoring of covalent adduct formation. While the carbonyl group of these compounds seemed to interact with the Trp_206_ side chain through hydrogen bonding, the P1′ substituent showed a different orientation. In compound **108**, the methyl group was directed towards Trp_206_, whereas in compound **109**, the groups were in proximity to residues Asn_173_, Val_152_, and Trp_206_, respectively, and the small distance between double bond and the sulfur atom indicating possible Michael addition. The removal of chlorine from the distal aryl moiety resulted in a derivative with reversible activity and low affinity for falcipain. Molecular dynamics studies suggested that the presence of chlorine may have contributed to a more stable conformation of compound **108** [76].

Previti et al. continued these lines of research by synthesizing a new series of peptide-based compounds as potential FP-2 and RD inhibitors, using compound **21** and k11002 (**110**) as lead compounds (Figure 33) [15]. The interaction of the compounds with FP-2 was lower than that presented for RD as cited previously in this article. They then tested several warheads and observed a trend of reactivity that followed this order: ketones > sulfones > esters > nitriles, with the vinyl ketones presenting the most potent inhibitors studied by Previti et al. (**111** and **112**; IC_50_ of 0.25 and 0.11 μM/EC_50_ of 3.16 ± 1.14 and 2.74 ± 1.02 μM *P. falciparum* respectively) (Figure 34).

##### Natural Compounds

With the purpose of taking advantage of natural compounds bearing privileged structures, Conroy et al. reported the synthesis and biological assay of gallinamide A [92] after its previous isolation [92]. From these efforts, the authors demonstrated interesting in vitro activity against *P. falciparum* with an IC_50_ of 50 nM of compound **90** and no harmful effect on red blood cells. They observed that replacing *N*-acyl pyrrolidine group had a negative impact on activity against FPs and parasites and assessed whether one or both olefinic moieties were necessary for the inhibition. They observed that the reduction in acyl pyrrolidine or in the α,β-unsaturated portion caused a detrimental effect in antiplasmodial activity.

Also, the substitution of *N*-acyl pyrrolidine with other C-terminal groups results in a loss of activity. The derivatives in Figure 35 presented interesting activity against *P. falciparum*. Substitution of *N*-acyl pyrrolidine was attempted with benzylamide, although it was not successful. They attributed the importance of this moiety to a Michael acceptor for the active-site Cys residues in FPs in comparison with the corresponding amide. Chain extension was also attempted but failed due to the loss of activity [92].

Considering both compounds **107** and **108,** Stoye et al. (Figure 34) proposed bioisosteric replacements of the ester bond for an amide bond in an attempt to stabilize the compounds metabolically [77]. Moreover, the authors maintained indole-derived pyrrolidone present in the gallinamide A, which possessed significant activity towards RD. Then, a target library was constructed, and the compounds were analyzed against FP1 and FP2 and R1 variation appeared to be well-tolerated. The half-lives of plasma and blood were increased by sterically hindered substituents in 1 and 3. Thus, they assessed how **117** acted against some strains of *Plasmodium falciparum*, being more potent than its previous leads and was evaluated against mice infected with *Plasmodium*. Further studies might be done for deeper conclusions.

#### 2.3.3. Proteasomes

Proteasomes are responsible for regulating specific processes within the cell cycle. Its composition consists of 20S catalytic units, which embrace 2 heptameric rings of beta subunits. The β1 unit behaves as caspase-like structures (ceasing itself after acidic residues), while the β2 subunit presents trypsin-like activity (cleaves after basic residues) and β5 subunit, chymotrypsin-like activity [80].

Proteasome inhibitors have potential activity against Malaria. Also, they have been discovered to be the cause of apoptosis in certain tumor-derived cells, which made their appearance in investigations against some types of cancer [80]. Furthermore, they can be active against artemisinin-sensitive and resistant parasites.

##### Benzoxaboroles

In past endeavors, epoxyketone and vinyl sulfone warhead-based inhibitors were synthesized and bound irreversibly to proteasome active sites. Bortezomib and ixazomib are two reversible boronate-based inhibitors used clinically to treat multiple myeloma. Bortezomib presents activity against *P. falciparum*, albeit its activity is not entirely linked to proteasome inhibition. In addition, boronic acid derivatives are known to be serine proteases covalent inhibitors [81] and, usually, boronate peptides are considered for their properties as covalent inhibitors as shown by Groll et al. [80]. For this reason, Xie et al. used direct evolution and screening from a boronate peptide library of human proteasome inhibitors from Takeda Oncology Company to identify inhibitors of the growth of *P. falciparum* in vitro [78].

Four compounds had good antiplasmodial activity (Figure 36), but compound **121** presented the best selectivity against β2 site comparing plasmodium and human structures (K_i_ (Pfβ2) = 6.4 nM/K_i_ (Hβ2) = 487 nM). Compound **120** presented similar t_½_ at the human β5c site (110 min) concerning dissociation from β5 site (106 min, K_i_ = 0.1 nM], but only 2 min (K_i_ = 490 nM) at the B2c site, indicating its preference for β5c site. As opposed to it, **120** had an interesting affinity against β2 and β5 sites of Pf20S (~5 nM) and longer t_1/2_ (~65 min). After crystallographic assays with pf20s and bortezomib, a covalent bond between threonine and boronate was observed (Figure 36). The boronic acid derivative moiety gave hard-atom oxygen a good spot to react with as Lewis hard-soft theory states. In addition, the amine present in the threonine performed hydrogen interactions with hydroxyl from boronate, and the Gly_47_ also performed this interaction (Figure 36).

Following these studies, Xie and collaborators attempted to enhance the oral bioavailability of MPI-1 (compound **118**) and synthesized a series of compounds with a single amide-bond backbone [78,79]. As a result, they obtained **122** (Figure 36)**,** an orally bioavailable antimalarial proteasome inhibitor with in vivo efficacy. As drawbacks and points of improvement, it was observed enhancement of in vivo half-life and possible inhibition of human T cell activation. Nevertheless, this series demonstrated the potential for a proteasome inhibitor to be used for Malaria in treatment, transmission-blocking, and chemoprophylaxis scenarios, especially when resistance to artemisinin limits the efficacy of existing combinations.

In a cryo-electron microscopy (EM) experiment, the same authors observed interesting contacts between **121** and the active site of Pf20S (Figure 35). The continuous density observed between Thr_1_ and boronate derived structures was the main feature observed. Another point to highlight was the great number of hydrophobic interactions, particularly near the biphenyl ring.

As it was observed in these assays, the size of the P1 substituent was directly related to potency and selectivity. When phenyl was replaced by biphenyl, **121** was a 5 nM inhibitor of Pfβ5 and a 21-nM inhibitor of 3d7 culture growth. As it was observed, this series presented favorable physico-chemical properties. For further studies, **121** and **122** were assessed for in vivo efficacy. Also, cryo-EM was performed in combination with MPI-5 and bortezomib. From this structure, it could be inferred that MPI-5 performed mainly hydrophobic interactions, especially considering biphenyl moiety, which probably accounted for the gain in potency in comparison with derivatives that only present monophenyl structure.

Also, from further characterization, it was observed that the covalent bonding to bortezomib is 17 times faster that the one characterized by **121** for HsB5c, while the latter residence time is nearly half of the bortezomib in Hs20s. As opposed to this, the rate of dative bond formation on Pf20S is higher than for bortezomib and similar for bortezomib and Hs20 beta 5C, which is consistent with a 14-fold higher inhibition of bortezomib with Hs20s beta 5c and 2.2-fold less potent pf20s beta5 in comparison to **121**. In conclusion, they could infer that **122** is a good example of orally bioavailable antimalarial compound presenting in vivo efficacy. However, it would be interesting to improve its half-life in vivo further.

In a separate study, Li et al. used liquid chromatography tandem mass spectrometry to screen 228 synthetic tetradecapeptides [101]. This resulted in **124**, in which they discovered a density continuity between this compound and two of *P. falciparum* 20S, as well as significant selectivity over the human enzyme, and **123** (Figure 37). These compounds explore bulky SAR in many positions in β2 and β5 subunit active sites of the proteasome.

As a continuation of this study, Stokers et al. explored whether **123** and **124** were interesting compounds against *P. falciparum* resistant strains (Figure 38) [82]. They observed that both had nanomolar potency against diverse parasites. Also, they would state that both of them seem to be more active against the schizont form of the parasite and early ring stages, which is interesting as most of the antimalarial compounds had been focused on the trophozoite stage.

##### Fk506-Binding Protein 35 (PfFKBP35)

The recruitment of PfFKBP35 is essential for immunosuppression carried out by inhibitors of calcineurin and mTOR [88]. Bianchin et al. offered insights through crystallographic studies of the interactions of PfFKBP-rapamycin [89]. They mentioned the differences between *Plasmodium* and human FKBP, highlighting the presence of Cys_106_ and Ser_109_ instead of His_87_ and Ile_90_ in humans (Figure 39). As a result, they confirmed that Cys_106_ might be investigated as a particular target for covalent inhibition, which was then done through QM/MM research by [90].

In studies carried out by Atack et al., similar antiplasmodial activity was observed between FK506 (compound **125**) and rapamycin (compound **126**) [91]. Two interesting compounds were obtained, **127** and **128**, presenting activity similar to that of rapamycin (1.4 and 1.9 vs 1.1 μM of IC_50_, respectively) in addition to the low cytotoxicity in HEK293T cell assays (Figure 40).

## 3. Concluding Remarks and Perspectives

Many new methodologies have been studied in the last few years concerning the advent of new covalent inhibitors.

DEL libraries, which consist of compounds tethered to unique DNA sequences serving as barcodes, are constructed and utilized in a combinatorial “split-and-mix” strategy to generate a large number of compounds [102]. The flow of a DEL involves library construction, selection against biological systems, and library decoding and hit picking. Recently, this technique has led to the discovery of irreversible binders, particularly in covalent DELs where all compounds are terminated in electrophilic warheads, enabling affinity selection for covalent hit discovery [103]. We believe that this can be used in neglected diseases niche, speeding the search for new drugs, with the advent of new DEL libraries containing compounds tested in these.

In addition, the use of fragment libraries has gained attraction in the field of covalent drug discovery [104]. These libraries are based on the knowledge and experience, and it is crucial to predict the reactivity of potential library members. Various methods, such as calculating Hammet sigma values of aromatic compounds and comparing the impact of individual substituents, or using indices and reactivity descriptors, can be employed for this purpose, especially concerning cysteine protease inhibitors in neglected diseases.

The compilation of a designed set of covalent fragments contributes to fragment reactivity and specificity, and both covalent fragments and libraries have become integral components of drug discovery approaches [104]. To expand the range of libraries, attention has been directed towards new labeling reactions and new warheads beyond the commonly used acrylamides [104].

At last, in parallel with the emergence of alphafold, novel chemoproteomics methodologies have gained significance [105]. For example, in the development of selective inhibitors for individual DGKs, the discovery of protein pockets available for inhibitor binding in the cellular environment can greatly benefit from diacylglycerol studies [105].

In this review, we have highlighted the significant challenges posed by neglected diseases, also known as diseases of poverty. We have specifically focused on three of the most crucial of them and highlighted the need for new drugs due to the limited number of available chemotherapeutic agents. To address this issue, researchers have been exploring new targets and new mechanisms of action for bioactive compounds. Covalent enzyme inhibitors, which have primarily been studied in cancer research, are gaining increasing interest for other diseases as well. This interest could be instrumental in developing new and improved chemotherapeutic agents for neglected diseases, which affect over one billion people worldwide.

Although we have primarily focused on sleeping sickness, Chagas disease, and Malaria, we have discussed the limited research on covalent inhibitors for neglected diseases in general. Few studies have been conducted on covalent inhibition, and the testing parameters are largely restricted to phenotypic assays. As such, there is no well-known gold standard or pattern recognized by the scientific community for neglected diseases concerning covalent inhibitors.

Considering covalent inhibitors have regained their importance in the last decades [106], we believe that exploring this mechanism of enzyme inhibition for neglected diseases could potentially validate phenotypic studies. In addition, using rational design tools in a genetic approach for identification of new bioactive molecular structures from these phenotypic assays might help scientists to improve the scarce chemotherapy for most of those diseases.

We have presented several compounds that have been studied and could serve as scaffolds for new compounds that may be explored for these three neglected diseases and others. We hope this review will inspire medicinal chemists who work in this area or who wish to use this approach in the future. We wholeheartedly agree with DNDi’s statement that “nobody should suffer from the lack of suitable treatments because their diseases are not profitable” [107]. It is imperative to address the issue of neglected diseases and develop new and effective treatments to ensure that people living in poverty have access to the same quality of healthcare as those in developed nations.

## Data Availability

Data is contained within the article.

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
