# Peer review of "Covalent Inhibitors for Neglected Diseases: An Exploration of Novel Therapeutic Options"

_pharmaceuticals, 2023, doi:10.3390/ph16071028_

Round 1
Reviewer 1 Report
The review manuscript titled “Covalent Inhibitors for Neglected Diseases: An Exploration of Novel Therapeutic Options” by Erick Tavares Marcelino Alves et al covers development of covalent inhibitors targeting neglected diseases.
While the manuscript can provide representative examples of cysteine targeting covalent inhibitors, no other residues except for proteasome inhibitor targeting threonine were discussed. There are a number of points that need the author's attention as shown below.
In Fig 2, the structures can be marked with the circle for the site of covalent modification. The structure of compound 6 should be checked (no SH and no tertiary amine) and it is unclear how compound 5 and 7 can be listed as a covalent inhibitor. The text should be clearly rewritten if they are covalent inhibitors or just hit molecules.
In page 6, line 1 , the scientific notation of K2nd should be corrected. 67000 ± 432 x 103 M-1 min-1 is not correct.
In page 6, line 197 , there is an error in the text (Kia Ki of 0.51 ± 0.36 nM)
In Fig 3, please check d) where the thiol attached to a wrong carbon. In addition, indicate the stereochemistry of the peptides along with their 3-dimentional structures for the docking poses rather than 2-D ligand interaction map.
In Fig 4, again be consistent with scientific notaion for Kcat/Ki.
In Fig 8, please fix the structure of compound 38.
In Fig 10, is the compound name is compound 38 or 30? The figure legend and the text do not match.
In Fig 17a, do the compound 56 and 57 contain CN attached to the sp3 carbon or aromatic?
In page20, line 442. The heading (3.1.3.4) was wrong.
Fig 19 is in conflict with Fig 20 in the direction and it is hard to understand what P1, P2, P3 meant. Please removed fig 19 and rework on Fig 20.
Fig 20, the table can be incorporated in the figure or Ki or Ic50 can be presented preferably under the compound number.
Fig 22, please show 3D docking pose rather than 2D-ligand interaction map for docking.
Fig 30 does not fully provide a useful information for the readers.
Fig 32 can be redrawn from the PDB to indicate the interaction of the ligand in the zoomed inset.
Fig 35 legend can be rewritten.
Overall, this review is okay to publish if the fine tuning is made for the figures.
Author Response
REVIEWER #1
The review manuscript titled “Covalent Inhibitors for Neglected Diseases: An Exploration of Novel Therapeutic Options” by Erick Tavares Marcelino Alves et al covers development of covalent inhibitors targeting neglected diseases.
While the manuscript can provide representative examples of cysteine targeting covalent inhibitors, no other residues except for proteasome inhibitor targeting threonine were discussed. There are a number of points that need the author's attention as shown below.
We would like to thank the reviewer #1 for the careful reading, and mention that the text has now been extensively reviewed and all novel-modified parts are highlighted in green.
Q1: In Fig 2, the structures can be marked with the circle for the site of covalent modification. The structure of compound 6 should be checked (no SH and no tertiary amine) and it is unclear how compound 5 and 7 can be listed as a covalent inhibitor. The text should be clearly rewritten if they are covalent inhibitors or just hit molecules.
Answer: It is true that it is unclear that compounds 5 and 7 are covalent inhibitors.
Q2: In page 6, line 1, the scientific notation of K2nd should be corrected. 67000 ± 432 x 103 M-1 min-1 is not correct.
Answer: We however did not understand why the notation is not correct. Should it be: 67000 ± 432 x 103/ M. min? We have taken this notation from the reference.
Q3: In page 6, line 197 , there is an error in the text (Kia Ki of 0.51 ± 0.36 nM)
Answer: Hence, we corrected : with a Ki of 0.51 ± 0.36 nM and EC50 of 1.65 ± 0.07 μM.
Q4: In Fig 3, please check d) where the thiol attached to a wrong carbon. In addition, indicate the stereochemistry of the peptides along with their 3-dimentional structures for the docking poses rather than 2-D ligand interaction map.
Answer: We also put as letter e the docking pose taken from the article.
Q5: In Fig 4, again be consistent with scientific notaion for Kcat/Ki. TIRAR ESSA COR
Answer: Sorry, but we did not identify kcat/ki in fig 4.
Q6: In Fig 8, please fix the structure of compound 38.
Answer: We performed the correction of compound 38.
Q7: In Fig 10, is the compound name is compound 38 or 30? The figure legend and the text do not match.
Answer: The compound related to the figure is number 35. It is corrected in the current version.
Q8: In Fig 17a, do the compound 56 and 57 contain CN attached to the sp3 carbon or aromatic?
Answe. Thanks for noticing, we already fixed it. Please, see figure 17 in the revised version.
Q9: In page20, line 442. The heading (3.1.3.4) was wrong.
Answer: The heading was corrected to 2.1.3.4.
Q10 : Fig 19 is in conflict with Fig 20 in the direction and it is hard to understand what P1, P2, P3 meant. Please removed fig 19 and rework on Fig 20.
Answer: Figure 19 was removed and p1,p2 and p3 were pointed out in figure 20.
Q11 : Fig 20, the table can be incorporated in the figure or Ki or Ic50 can be presented preferably under the compound number.
Answer: The values were presented under the compound number.
Q12 : Fig 22, please show 3D docking pose rather than 2D-ligand interaction map for docking.
Answer: The 3d docking posing is presented in the corrected version.
Q13 : Fig 30 does not fully provide a useful information for the readers.
Answer: We accepted the recommendation and Fig30 was, then, removed.
Q14 : Fig 32 can be redrawn from the PDB to indicate the interaction of the ligand in the zoomed inset.
Answer: The carbon has been attached to tiol.
Q15: Fig 35 legend can be rewritten.
Answer: Caption was rewritten as required.
Overall, this review is okay to publish if the fine tuning is made for the figures.
Reviewer 2 Report
Authors compiled a valuable and comprehensive review of the field of covalent inhibitors targeting proteins responsible for rare and so called neglected diseases. There are a few modifications suggested before accepting the paper for publication.
1. Table 1 does not summarize recent covalent inhibitors, but targets and corresponding references. Either the caption and the referring text or the Table should be changed.
2. Covalent binding is a two step procedure where the non-covalent skeleton forms the first interactions. This might be included to clear the mechanism of action of a covalent binder.
3. Ki might be defined as binding affinity.
4. On the ligand interaction figures the charges and valences are suggested to be checked. E.g. is Fig 2 (b) correct?
5. On Fig 2 parameters observed are not shown.
6. In general, tables contain Ki Kinact etc. Would it be possible to add IC/EC50s as well? Moreover, most tables seems sometimes empty, as there are many columns, but only a few data. Those data could be added to the figures, and then space could be saved.
7. In general, the paper is very long. In my view the presentation of SARs, many substiuents is less important from side of covalent inhibitors, so I suggest to include only the best/most significant derivatives from the same family with the same warhead. Of course, in case of multiple warheads, multiple compounds could be included.
8. If a "library" is mentioned, it is sugested to include the number of compounds, and probably a hit rate if available.
9. Fig17c shows ligand interactions and not a crystal structure contrary to the caption.
10. It is suggested to revise how many interaction figures needs to be presented. E.g. Fig 25 does not seem very important. Fig 32 is a nice figure, but is it really needed? The ligand interactions are informative in case of showing covalent binding mechanism, and it might be considered in some cases to show the full stucture of the protein with the ligand in the pocket.
11. Sometimes the figures are too far from the corresponding text.
11.
Author Response
REVIEWER#2
Authors compiled a valuable and comprehensive review of the field of covalent inhibitors targeting proteins responsible for rare and so called neglected diseases. There are a few modifications suggested before accepting the paper for publication.
We would like to thank the reviewer #2 for the careful reading, and mention that the text has now been extensively reviewed and all novel-modified parts are highlighted in blue.
Q1: Table 1 does not summarize recent covalent inhibitors, but targets and corresponding references. Either the caption and the referring text or the Table should be changed.
Answer: The caption was changed from “Covalent inhibitors studied on three of the most important Neglected diseases” to “targets and references containing potential covalent inhibitors on three of the most important Neglected diseases”.
Q2: Covalent binding is a two-step procedure where the non-covalent skeleton forms the first interactions. This might be included to clear the mechanism of action of a covalent binder.
Answer: The part of the text that stress this out is present in Covalent inhibitors for neglected diseases section as: It is important to stress that the covalent inhibition is a two-step procedure as pictured on scheme 1. Some of the constants involved in the process are described below.
Q3: Ki might be defined as binding affinity.
Answer. In the definition of Ki, we also stated this was related to binding affinity.
Q4: On the ligand interaction figures the charges and valences are suggested to be checked. E.g. is Fig 2 (b) correct?
Answer: We believe that figure 2(b) is correct, but we carefully looked throughout all the structures and corrected possible structure errors such as the one related to figure 2 (a) compound 6.
Q5: On Fig 2 parameters observed are not shown.
Answer: The parameters are now shown in a table below the molecules in Fig 2.
Q6: In general, tables contain Ki Kinact etc. Would it be possible to add IC/EC50s as well? Moreover, most tables seems sometimes empty, as there are many columns, but only a few data. Those data could be added to the figures, and then space could be saved.
Answer: Sometimes values for IC and EC50 has not been shown in the references.
Q7: In general, the paper is very long. In my view the presentation of SARs, many substiuents is less important from side of covalent inhibitors, so I suggest to include only the best/most significant derivatives from the same family with the same warhead. Of course, in case of multiple warheads, multiple compounds could be included.
Answer: There are much information about the topic in the literature. That is why it is so long.
Q8:. If a "library" is mentioned, it is sugested to include the number of compounds, and probably a hit rate if available.
Answer: We have now mentioned the number of compounds in each one of the libraries presented in the text. Hit rate, however, was not available for most of them.
Q9: Fig17c shows ligand interactions and not a crystal structure contrary to the caption.
Answer: Caption has been rewritten.
Q10: 10. It is suggested to revise how many interaction figures needs to be presented. E.g. Fig 25 does not seem very important. Fig 32 is a nice figure, but is it really needed? The ligand interactions are informative in case of showing covalent binding mechanism, and it might be considered in some cases to show the full stucture of the protein with the ligand in the pocket.
Answer: Figure 25 was removed and Fig32 was remade.
Q11: Sometimes the figures are too far from the corresponding text.
Answer: We tried to make the figures closer to their corresponding text
Round 2
Reviewer 2 Report
Authors answered all the questions and comments, however, there are still some of those that are suggested to be modified.
On Fig 2(b) nitrogen atoms are missing from molecule 6 and the double bond of thiophene is different on Figs 2a and 2b.
I still believe that tables like Fig 9 are not necessary, it is almost empty, it would be better to have all those data on the Figures.
I also believe that the manuscript should be shorter. E.g. Fig 27 contains many similar structures that differ only in the substituents. There might be no need to show all the almost same structures. I suggest to shorten the manuscript and focus the Figures.
Author Response
Dear editor
Please, find enclosed the revised version (Round2) of our manuscript entitled “Covalent Inhibitors for Neglected Diseases: An Exploration of Novel Therapeutic Options (2012 - present)” by Erick Tavares Marcelino Alves, Filipe Gomes Pernichelli, Lucas Adriano do Nascimento, Glaucio Monteiro Ferreira and Elizabeth Igne Ferreira.
We hope this revised version could be considered for publication in the Pharmaceuticals. We think the comments have been insightful, and our work has greatly improved after implementing the reviewers’ comments.
We are submitting the revised manuscript with all the changes highlighted. Additionally, we have responded to all comments.
Sincerely yours,
Elizabeth Igne Ferreira
Reviewer #2
The authors answered all the questions and comments, however, there are still some of those that are suggested to be modified.
General comments
We would like to thank the reviewer for the careful reading and mention that the text has now been extensively reviewed and all novel-modified parts are highlighted in yellow.
Q1. On Fig 2(b) nitrogen atoms are missing from molecule 6 and the double bond of thiophene is different on Figs 2a and 2b.
Answer: The figure now was corrected.
Q2. I still believe that tables like Fig 9 are not necessary, it is almost empty, and it would be better to have all those data on the Figures.
Answer: We agreed. Now the figure has the data regarding every molecule.
Q3. I also believe that the manuscript should be shorter. E.g. Fig 27 contains many similar structures that differ only in the substituents. There might be no need to show all the almost same structures. I suggest to shorten the manuscript and focus the Figures.
Answer: We agreed. 20 compounds have been removed from the article, diminishing the number of compounds from 145 to 126.